# How Are Information and Communication Technologies Supporting Routine Outcome Monitoring and Measurement-Based Care in Psychotherapy? A Systematic Review

**DOI:** 10.3390/ijerph17093170

**Published:** 2020-05-02

**Authors:** Patricia Gual-Montolio, Verónica Martínez-Borba, Juana María Bretón-López, Jorge Osma, Carlos Suso-Ribera

**Affiliations:** 1Department of Basic and Clinical Psychology and Psychobiology, Jaume I University, Avda. Vicent Sos Baynat s/n, 12071 Castellon de la Plana, Spain; al384421@uji.es (P.G.-M.); borba@uji.es (V.M.-B.); breton@uji.es (J.M.B.-L.); 2Department of Psychology and Sociology, Universidad de Zaragoza and Instituto de Investigación Sanitaria de Aragón, Ciudad Escolar s/n, 44003 Teruel, Spain; osma@unizar.es

**Keywords:** information and communication technologies, outcome monitoring, therapist feedback, measurement-based care, mental health

## Abstract

Psychotherapy has proven to be effective for a wide range of mental health problems. However, not all patients respond to the treatment as expected (not-on-track patients). Routine outcome monitoring (ROM) and measurement-based care (MBC), which consist of monitoring patients between appointments and using this data to guide the intervention, have been shown to be particularly useful for these not-on-track patients. Traditionally, though, ROM and MBC have been challenging, due to the difficulties associated with repeated monitoring of patients and providing real-time feedback to therapists. The use of information and communication technologies (ICTs) might help reduce these challenges. Therefore, we systematically reviewed evidence regarding the use of ICTs for ROM and MBC in face-to-face psychological interventions for mental health problems. The search included published and unpublished studies indexed in the electronic databases PubMed, PsycINFO, and SCOPUS. Main search terms were variations of the terms “psychological treatment”, “progress monitoring or measurement-based care”, and “technology”. Eighteen studies met eligibility criteria. In these, ICTs were frequently handheld technologies, such as smartphone apps, tablets, or laptops, which were involved in the whole process (assessment and feedback). Overall, the use of technology for ROM and MBC during psychological interventions was feasible and acceptable. In addition, the use of ICTs was found to be effective, particularly for not-on-track patients, which is consistent with similar non-ICT research. Given the heterogeneity of reviewed studies, more research and replication is needed to obtain robust findings with different technological solutions and to facilitate the generalization of findings to different mental health populations.

## 1. Introduction

The effectiveness of psychotherapy for the treatment of mental disorders has been supported by an impressive amount of evidence. However, some patients do not respond to treatment as expected, either because they do not show an improvement during the intervention or they discontinue it, or because they show a deterioration [1,2]. There might be several reasons explaining individual differences in response to psychological treatments, including unchallengeable patient characteristics (e.g., age), their personality and behavioral profiles, treatment characteristics and context, and patient physical health status and life context, to name some examples [3]. While acknowledging the previous, an increased number of studies have pointed to methodological deficits, namely, in how patients are monitored during treatments, as key factors influencing current treatment effectiveness [4,5].

Specifically, it has been proposed that a paradigm shift in the practice of psychotherapy towards an ecological momentary assessment (EMA) is necessary in order to monitor patients repeatedly and frequently in their natural environment [6,7]. However, simply monitoring the patient does not appear to be enough to improve the patient’s outcomes [8,9]. In this sense, therapist (and patient) feedback has been argued to be a fundamental aspect if patient monitoring during therapy is to be effective [1,10].

The aforementioned procedure is known by different terms in the literature, such as routine outcome monitoring, with outcome or continuous feedback; progress monitoring; or, probably the most popular, measurement-based care (MBC) [9,11]. For simplicity and readability, the latter will be preferred throughout this text.

MBC is defined as a periodic and recurrent assessment of the patient’s status over the course of an intervention using standardized measures. Importantly, the evaluation is followed by immediate, frequent, and systematic feedback of the patient’s information to the therapist [12,13]. This procedure has been argued to help therapists to assess actual patient progress, suggest necessary adjustments to the treatment, and identify patient deterioration or improvement trajectories, thus enhancing the patient’s response to the intervention [1,14,15,16]. According to the American Psychological Association’s (APA) Division 28 Task Force on Empirically Supported Relationships, MBC may also lead to an improvement in the therapeutic alliance and may avoid premature treatment termination, because MBC encourages collaboration between patients and therapists, thus promoting engagement, dialogue and discussion of real-life, daily patient difficulties during sessions. Furthermore, giving feedback to patients raises awareness of their progress and makes them become more mindful of their symptoms, which may also enhance the quality of psychological interventions and the patient–therapist alliance [14].

Several studies have shown the efficacy of this systematic patient monitoring with progress feedback to the therapist in psychological interventions [10,17]. Research also indicates that feedback from outcome measures enhances treatment effectiveness, particularly in not-on-track patients (i.e., those who do not make the expected changes) or when it is provided both to clinicians and patients [8,10,13,14,17,18]. Specifically, providing the therapists with immediate feedback about the patient’s symptoms appears to reduce the number of early dropouts and improve several treatment outcomes (depressive and anxiety symptoms, psychosocial functioning, psychosis, quality of life, therapeutic alliance, etc.) when compared against usual treatment [17,19]. Overall, medium effect sizes have been reported when using MBC [1,14].

What the existent literature suggests is that MBC is a promising methodology to enhance the effectiveness of psychological treatments. However, there are a number of flaws into the literature on MBC that might have limited the impact and dissemination of this procedure [20,21]. Traditionally, MBC has been conducted with self-report, paper-and-pencil questionnaires that patients complete before or after therapy sessions. Additionally, assessments are mostly retrospective and based on the patient’s recalled experiences during the past week [11]. With this information, the therapist examines and discusses the results during the actual therapy session [22]. As noted earlier, while this practice has been shown to provide some relevant information, relying on paper-and-pencil retrospective reports only, where daily experiences are not reported, might result in recall bias, thus making it difficult to understand patient fluctuations over time [23,24]. Furthermore, focusing on self-reports exclusively is problematic, as more objective data (e.g., actual number of steps taken or time spent out of the home) is ignored or based on patient appraisal only.

Currently, the rapid growth of new technologies in our society has changed the way psychotherapy is conducted. For example, information and communication technologies (ICT) have been argued to allow therapists to evaluate and receive patient progress feedback in real time, thus minimizing patient recall bias [25]. Additionally, the use of ICT allows obtaining objective data of patient changes in natural settings, for example, using sensors (accelerometers, positioning system, or pedometers, among others) [22]. Therefore, the use of handheld ICT devices such as smartphones, tablets, or laptops might increase the effectiveness of MBC by facilitating EMA before, during, and after treatment, providing the information immediately to therapists and researchers, and making it easier to combine collection of objective and subjective patient data [11,21,25].

In order to investigate to what extent ICTs are being implemented to enhance psychological interventions and how their application is effectively improving outcomes, we have conducted a systematic review to explore how ICT is being used for MBC in face-to-face psychological treatments. In doing so, we have investigated: 1. what the different technologies and procedures used for MBC during psychological interventions are and 2. to what extent the use of ICT for MBC is feasible, acceptable, and effective.

## 2. Materials and Methods

### 2.1. Search Strategy

The search was conducted in accordance with the Preferred Reporting Items for Systematic Reviews and Meta-Analyses [26]. The search was conducted in February 2020 and included published and unpublished studies from the electronic databases PubMed, PsycINFO, and SCOPUS. In addition, reference mining was performed by searching through bibliographies of relevant articles. The selection of these databases was motivated by previous research showing that PsycINFO has high sensitivity and specificity when retrieving intervention studies and is especially suitable for psychology research, and that SCOPUS offers about 20% more coverage than other important databases such as Web of Science. Additionally, it has been argued that Google Scholar provides inaccurate results, and PubMed is one of the preferred tools for biomedical research [27,28].

The search strategy included variations of the terms “psychological treatment”, “progress monitoring”, and “technology” (See Appendix A for the complete list of search terms and combinations). Due to the diversity of terms, a broad search strategy of terms was used. Synonyms, abbreviations, and spelling variations were identified for the three concepts and combined in the search using the “OR” Boolean operator, with non-synonymous concepts combined using “AND”. These terms were searched in titles and abstracts. The references of included studies and relevant systematic reviews were searched to identify studies that were missed during the literature search. There were no restrictions regarding language or publication period, but the search was only conducted in English.

### 2.2. Inclusion Criteria

Included studies were psychological treatments enhanced by MBC using technology systems. Specifically, included studies 1. were clinical trials (either feasibility, case studies, and both randomized or non-randomized investigations); 2. included the use of technology during MBC (both for monitoring and feedback provision) while undergoing a face-to-face psychological intervention; and 3. involved feedback to the therapist or to both therapists and patients based on standardized measures.

To be considered MBC, the intervention must satisfy the following components: 1. routine assessment of a symptom, an outcome, or a process measure; 2. therapist review of data; and 3. therapist use of data to inform clinical decisions. Therefore, the study population can include patients with any mental disorder from all ages who are routinely monitored via validated outcome measures using technologies over the course of a face-to-face psychological treatment. 

### 2.3. Exclusion Criteria

Studies in which technology systems were not used in the whole MBC process, including the assessment and feedback parts, or where only patients but not therapists were provided with feedback, were excluded. In addition, studies were excluded if they did not include a face-to-face psychological intervention.

### 2.4. Search and Screening 

Initially, 193 publications were identified from database searches and screening of reference lists (see Figure 1 for the study diagram flow). After excluding duplicates (*n* = 63), 130 publications were retained for screening. After initial screening of titles and abstracts, 84 of these documents were excluded due to eligibility reasons. For the remaining 46 publications, full texts were retrieved. After eligibility assessment of the full texts, 28 publications were excluded. The majority of publications were excluded because they did not meet the eligibility criteria, resulting in a final sample that comprised 18 publications. 

The search, screening process, and data extraction were conducted independently by the first two authors (PGM and VMB). When in doubt, study eligibility was discussed with a third author (CSR). After the phase of study eligibility assessment, inter-rater agreement was calculated (Cohen’s kappa). This coefficient showed a substantial overall agreement, represented by a kappa of 0.908 (*SD* = 0.064; 95% CI, 0.781, 1.000).

### 2.5. Data Extraction

The following data were extracted from each included study, using a standardized data-extraction form developed a priori: authors, study setting (geographical setting and type of clinic), sample size, study design, study participants (demographics and type of mental disorder), type of psychological intervention, assessment characteristics (frequency and setting), primary outcomes, feedback characteristics (to whom, frequency, and setting), type of technology used, technology feasibility, and clinical effectiveness. Data was extracted from all full texts by the first author (PGM) and then discussed with another author (VMB) before it was reviewed by all co-authors.

### 2.6. Risk of Bias Assessment

All studies included in this review were independently rated for quality by two reviewers (PGM and VMB). If the rating differed, reviewers discussed the articles to reach consensus with a third reviewer (CSR). The Study Quality Assessment Tools from the National Heart Lung and Blood Institute [29] were used to assess study quality and risk of bias. This tool was preferred because it includes six types of studies and specific criteria according to the study design (i.e., controlled intervention studies; systematic reviews and meta-analyses; observational cohort and cross-sectional studies; case-control studies; before–after studies with no control group; and case series studies). This tool allows reviewers to rate studies as “good”, “fair”, or “poor”. Total quality scores ranged from 9 to 14 points depending on the study design.

### 2.7. Synthesis of Results

Frequency tables were used to summarize the characteristics of individual studies. We conducted a systematic review and not a meta-analysis, because the emphasis was not on effect sizes, but on how MBC with technology was being conducted. Additionally, we anticipated that included studies would be very heterogeneous, because the review includes any form of psychological intervention, all types of technologies, several trial designs, and different types of mental disorders and outcomes. Thus, we performed a narrative synthesis only.

### 2.8. Additional Analyses

Factors affecting study heterogeneity included variations in the type of mental disorder (e.g., major depressive disorder or anxiety disorders), outcomes included, treatment characteristics (type, format, and duration), measures used (clinician-rated versus self-rated), study design, and differences in the means by which MBC was delivered. The description of the findings was sensitive about the aforementioned subgroups when possible.

## 3. Results

### 3.1. Characteristics of Included Studies

The characteristics of included studies are shown in Table 1. Of the 18 studies included in the systematic review, eight were published in the USA [30,31,32,33,34,35,36,37], with the remaining studies being published in Australia (*n* = 3 [38,39,40]), the United Kingdom (*n* = 3 [41,42,43]), Austria (*n* = 2 [44,45]), Greece (*n* = 1 [46]), and the Netherlands (*n* = 1 [47]). Most studies took place in outpatient settings (*n* = 15, 83.33%) such as mental health services (*n* = 3 [33,34,47]), specialist clinics (*n* = 4 [36,38,39,43]), hospital clinics (*n* = 7 [31,35,37,41,42,45,46]), and university clinics (*n* = 1 [32]); only three of them were conducted in inpatient settings [30,40,44]. In terms of design, six studies were feasibility pilot investigations (single group) [30,31,34,36,37,44]; four studies were case studies [32,39,45,46]; four studies were randomized controlled trials (RCTs) [35,38,42,47]; two of them were non-randomized controlled trials [33,40]; and two were pre–post investigations [41,43]. The sample sizes of the included investigations ranged from 1 to 2233 participants. 

The included studies targeted very heterogeneous diagnoses. However, depression and anxiety disorders were the most frequent (*n* = 6, 33.33% of studies [31,33,37,40,41,42]). The remaining disorders were schizophrenia spectrum disorders (*n* = 4, 22.22% of studies [34,36,38,39]), bulimia nervosa (*n* = 1, 5.6% of studies [45]), substance use (*n* = 1, 5.6% of studies [35]), dementia (*n* = 1, 5.6% of studies [46]), post-traumatic stress (*n* = 1, 5.6% of studies [30]), bipolar disorder (*n* = 1, 5.6% of studies [43]), couple problems (*n* = 1, 5.6% of studies [32]), and combinations of heterogeneous disorders together (*n* = 2, 11.11% of studies [44,47]). The majority of treatments were addressed to adult populations (*n* = 16) and only two investigations (*n* = 2 [34,36]) aimed at treating mental health problems in younger populations (i.e., adolescents and young adults).

Regarding the treatments offered in the included studies, different face-to-face interventions were provided across studies. However, cognitive behavioral therapy (CBT) was the most frequent (*n* = 13). Other therapeutic options were client-centered psychotherapy (*n* = 1 [45]), collaborative care (*n* = 1 [37]), an early psychosis program (*n* = 2 [34,36]), and couple multisystemic psychotherapy (*n* = 1 [32]). The format of the intervention was mainly individual (*n* = 15), but some studies included group treatments (*n* = 3 [30,33,40]). Finally, the intensity (i.e., frequency) of the intervention also differed across investigations. Some studies implemented a low-intensity treatment plan (i.e., less than eight sessions; *n* = 6 [32,35,38,39,43,45]), while others applied a higher-intensity intervention (i.e., more than eight sessions; *n* = [30,33,34,36,37,40,44,46]) or both a low- and a high-intensity treatment (*n* = 4 [31,41,42,47]).

### 3.2. MBC Characteristics 

#### 3.2.1. Assessment Procedure Used to Track the Patient’s Status

The characteristics of studies included in the review are described in Table 2. Most studies (*n* = 9 [30,31,33,35,38,39,40,44,46]) monitored their patients daily. The remaining studies monitored their patients weekly (*n* = 4 [32,41,42,47]), before every therapy session, or both daily and weekly (*n* = 5 [34,36,37,43,45]). In the latter studies conducting assessments both daily and weekly, daily assessments usually included shorter questionnaires that evaluated therapeutic process outcomes, mood, or medication adherence, while longer outcome scales (i.e., measures of depressive or anxiety symptoms) were administered weekly. Treatment effectiveness was assessed most commonly with the primary outcome measures of interest for the investigation, which most often were the Patient Health Questionnaire-9 for depressive symptoms and the Generalised Anxiety Disorder-7 for anxiety symptoms (*n* = 4 [33,37,41,42]). Other outcomes included the frequency and amount of drug use (*n* = 1 [35]); the Subjective Experiences of Psychosis Scale, the Auditory Hallucinations subscale of the Psychotic Symptom Rating Scales for psychotic symptoms, and the Depression Anxiety Stress Scale for negative emotional symptoms (*n* = 2 [38,39]); the Outcome Questionnaire-45 (*n*= 1 [47]); the Therapy Process Questionnaire (*n* = 1 [44]); the World Health Organization’s Wellbeing Index (*n* = 1 [40]) for well-being; the Symptom Checklist-90 for bulimia symptoms and the Intersession Experience Questionnaire for the psychotherapy process (*n* = 1 [45]); the Altman Self-Rating Mania scale and the 16-item Quick Inventory of Depressive Symptoms-Self Report scale (*n* = 1 [43]); and other clinical measures related to sleep, mood, medication use, or daily functioning (*n* = 6 [30,31,32,34,36,46]).

#### 3.2.2. Feedback Procedure

The majority of studies provided feedback to both therapists and patients (*n* = 15). The remaining studies gave feedback to therapists only (*n* = 3 [32,36,44]). Most studies used feedback to track the patients’ progress focusing on key aspects during treatment and to monitor responses in-between sessions. Even when the feedback was not directly provided to the patient, this information was used to discuss patient progress during treatment sessions or to take clinical decisions (e.g., emphasize a specific content during session).

Feedback included information about treatment evolution in progress charts, summary sheets, graphs of scores and curves, and plots of scores within trajectories. Feedback information was either sent periodically to the therapist in response to patient assessments or on-demand (weekly or daily). In some studies, feedback to the therapist appeared when the patients’ responses were considered clinically significant according to pre-established criteria.

### 3.3. Technology Characteristics 

The technology characteristics of included studies are presented in Table 2. Most studies included handheld technology, such as smartphone apps (*n* = 7 [31,34,36,37,38,39,45]), touch-screen technologies (*n* = 2 [40,46]), or laptops (*n* = 5 [32,41,42,44,47]), together with a web-based platform for the therapist. The remaining studies used phone text messages (*n* = 2 [30,33]), e-mail (*n* = 1 [43]), or a phone interactive voice response system (*n* = 1 [35]). For example, in one study, automated text messages and a web-based platform (e.g., HealthySMS) were used [33]. In other investigations, authors implemented an Internet-based system, such as the Patient Case Management Information System (PCMIS), or an electronic clinical record system which includes outcome-monitoring graphs that chart depression and anxiety scores at every session [41,42]. A similar example was the Synergetic Navigation System, an Internet-based device for data collection (with web-compatible devices such as PCs, tablets, or smartphones) and data analysis that allows for the implementation of questionnaires at any chosen interval [44]. Another Internet-based system was the DynAMo web app, a piece of software that combines algorithm-based treatment planning, process monitoring, and outcome monitoring, which can be used by both researchers and clinicians to plan treatments and monitor psychotherapeutic processes [45]. One investigation used the Systemic Therapy Inventory of Change System, an online system that assesses and tracks changes in the patients’ interpersonal system, as well as in the therapeutic alliance, and also feeds these data back to the therapists on demand [32]. A final example of an Internet-based system used in one of the included investigations was Ginger.io, an mHealth software program comprising a therapist dashboard and an app which can collect data from self-report surveys sent to the participant in addition to “passive” data from the participant’s phone, such as number of calls and SMS messages, and movement patterns based on Global Positioning System data [36].

#### 3.3.1. Technology Feasibility

Because of the large differences in sample sizes across investigations (Table 3), case studies with one to four participants [32,39,45,46] have been described in previous sections but will not be considered in the feasibility and effectiveness summaries. The remaining studies included at least 17 participants, and therefore will be discussed in detail here and in Table 3. Overall, the results demonstrated that enhancing MBC in psychological therapy with technology was generally feasible and acceptable (Table 3). This statement is supported by the high average response rate for daily (mean = 63.3%, range = 40%–81% [34,36,43]), weekly (mean = 73.0%, range = 39%–88% [34,36,37,43]), and monthly (92.7% [30]) symptom monitoring across studies, low average missing data rates amongst patients (13% [44]), and high completion rates (mean = 77.8%; range = 64.1%–90% [31,35,38,40,44]). In addition, several studies reported satisfaction with the technology used to improve MBC in the intervention delivered, and most of them revealed that patients and therapists would recommend the technology used as part of the treatment [31,34,38].

#### 3.3.2. Clinical Effectiveness

As summarized in Table 3, data about clinical effectiveness was described in seven studies, including four RCTs [35,38,42,47], two nRCTs [33,40], and a quasi-experimental pre–post investigation [41]. All RCTs included active controls (i.e., traditional face-to-face psychological treatment) without MBC. According to these investigations, the use of technology-supported MBC appears to significantly reduce symptom severity, and changes are sometimes larger than with traditional interventions [38], especially in patients at risk of poor response to treatment (i.e., not-on-track cases) [35,40,42,47], which is consistent with previous research [42,46]. Furthermore, one study showed that although traditional CBT and technology-enhanced MBC CBT were comparable in terms of treatment effectiveness, the latter significantly reduced therapy duration and cost of treatment [41]. Also in favor of technology-assisted MBC, another investigation revealed that patients stayed in therapy longer (i.e., higher adherence) in the experimental condition, that is, when MBC was supported by technology (group CBT with a text messaging adjunct) as opposed to traditional MBC without technology (group CBT without the text messaging adjunct) [33]. In sum, most studies suggest that technology-supported MBC has the potential to improve the efficacy and cost-effectiveness of psychotherapy, especially for not-on-track individuals.

### 3.4. Risk of Bias Assessment

As observed in Table 4, Table 5, Table 6 and Table 7, studies included in this review could be placed in four of the categories proposed by the National Heart Lung and Blood Institute [29], namely, case studies, before-after studies; observational cohort and cross-sectional studies; and controlled intervention studies. Overall, the four studies classified as case studies had a “good” quality, with total scores ranging from 5 to 7 points out of a maximum of 9 points [32,39,45,46]. Even so, none of them had follow-up sessions, or these were not reported, and two studies did not include a complete case definition [32,45]. Secondly, the two before–after studies could also be rated as “good” quality investigations, as both obtained 9 points of a maximum of 12 [41,43]. The main issue with one of the studies was related to sample size, although this concern was justified, as this was a pilot study and authors reported the previous in the limitation section [43]. The six studies classified as observational, cohort and cross-sectional studies correspond to feasibility and acceptability studies, and some quality criteria, such as numbers 7, 8, 12, and 14, were not applicable [30,31,34,36,37,44]. Overall, feasibility and acceptability studies did not meet most criteria required in observational and cross-sectional studies (they met only seven or eight criteria of a maximum of 14), so their quality could only be rated as “fair”. It is important to note that just two studies maintained 80% of the sample [30,31] and four studies did not meet the participation rate of 50% from eligible participant criteria [30,31,37,44]. Finally, two of the controlled intervention studies [33,40] were rated as “poor” quality as they were non-randomized and did not follow most of the criteria for controlled studies (i.e., randomization, blind allocation, or assessment). These two studies met, respectively, only two [40] and eight [33] criteria of a maximum of 14. The remaining four controlled intervention studies [35,38,42,47] were “good” quality investigations despite the lack of blinded allocation and assessment [35,42,47] and relatively high drop-out rates [42,47].

## 4. Discussion

The aim of the present study was to systematically review evidence on how ICT is being used for MBC in psychological treatments. To the best of our knowledge, this is the first attempt to systematically examine the different technologies that have been used so far for MBC during psychological interventions and to explore to what extent the use of ICT for MBC is feasible, acceptable, and effective.

One important finding was that only 18 studies met our inclusion criteria, which suggests that this is a field that requires more research. Additionally, the included studies varied greatly in terms of study design, diagnoses, MBC characteristics, and technology used, which again suggests that more investigation and replication will be needed to obtain robust findings about different technological solutions for MBC and to facilitate the generalization of the results to different populations. Future research should also examine the implication of technology systems for MBC in specific populations (e.g., children and adolescents, who are more familiar with technology systems). Furthermore, most of the included studies focused on mood and anxiety disorders, so it would be interesting to investigate the effects of technology used for MBC in other mental disorders.

As noted in the results section, the included technologies to support MBC were frequently handheld ICTs, such as smartphone apps, tablets, or laptops, which were involved both in the patient monitoring process and in the feedback to the therapists. In this sense, while tablets and laptops might be more difficult to use for EMA, it is encouraging that smartphone apps, which might facilitate EMA to a greater extent than other technologies, are also being used as supporting technologies for MBC.

An important finding regarding technology was that, overall, the use of technology for MBC during psychological interventions appears to be feasible and acceptable. In addition, technology in MBC was found to be effective [38] and cost-effective [41], particularly for not-on-track patients [35,40,42,47], as revealed in previous studies using MBC without technology [1]. Importantly, treatment engagement (i.e., time until dropout) was also enhanced with technology-supported MBC [33]. While these findings should be interpreted with caution, due to the limited number of existing investigations and the reduced number of controlled trials comparing technology-supported and non-supported MBC, the results suggest that the use of ICT to support MBC should continue to be tested in the future.

It is important to acknowledge that technology-supported EMA for MBC is different to a similar concept, which is ecological momentary intervention (EMI). Specifically, while in EMI a given intervention is provided in response to EMA in a timely manner (e.g., providing therapeutic skills with an app based on patient responses), in MBC EMA is only used to enhance face-to-face psychological interventions.

It is also important to note that some studies were excluded from this systematic review for several reasons which should be mentioned here. For instance, in some investigations, the monitoring process was not used to guide therapeutic decisions, even though technology was used for monitoring (thus, this would not be considered MBC). Conversely, in other studies, the technology was only used in one part of the MBC process, most frequently during the feedback part, but not for EMA (e.g., assessments were made in a paper-and-pencil approach, but then the information was introduced in a computer and presented in graphs or charts to the therapists and/or patients) [16,48,49,50].

### 4.1. Limitations

Some limitations should also be considered when interpreting the results of the present systematic review. As in previous similar reviews [13,18], the heterogeneity of studies with respect to sample size, measures used, and methodology, to name some examples, made it difficult to piece together the results and restricted the implementation of a meta-analysis which affects the generalizability and robustness of findings. Moreover, the majority of included studies were feasibility pilot studies, case studies, or non-RCTs. Although these designs can yield valuable information, RCTs, which have been rarer, are considered superior because of their higher internal validity and, therefore, higher robustness of the evidence indicating a (causal) relationship. Additionally, some factors might have biased the present systematic review findings, including the fact that only three databases were used for the search, and the possibility that studies where ICT was not feasible or did not add any value have not been published. Finally, it is important to note that this systematic review is limited to the interpretations of the authors who conducted the systematic review.

### 4.2. Conclusions

To conclude, this systematic review found preliminary support for the use of technology in MBC during psychological interventions. The use of ICTs in MBC has brought some encouraging contributions to the evolution of psychotherapy and its inclusion in routine care might significantly change the way psychotherapists work. Particularly, the provision of real-time information on symptom progress over the course of psychological interventions might help therapists detect and rapidly react to problems that might occur during treatment (e.g., exacerbation in symptomatology or low adherence to recommended practices). This would make current interventions more flexible and personalized [22] and should favor the psychotherapeutic relationship during face-to-face interventions. Additionally, this might increase the patients’ awareness of their own progress.

In addition, technology was generally found to be a feasible and acceptable add-on tool for the MBC process. Therefore, the use of technology for improving the MBC process is mostly supported, as it might facilitate EMA and offer some potential for improving psychotherapy thanks to the real-time connection between patient assessment and therapist and patient feedback [32]. While the presented results are, overall, encouraging, especially for not-on-track patients, more research is required in this field, especially RCTs comparing technology-supported MBC with traditional MBC without technology.

## Figures and Tables

**Figure 1 ijerph-17-03170-f001:**
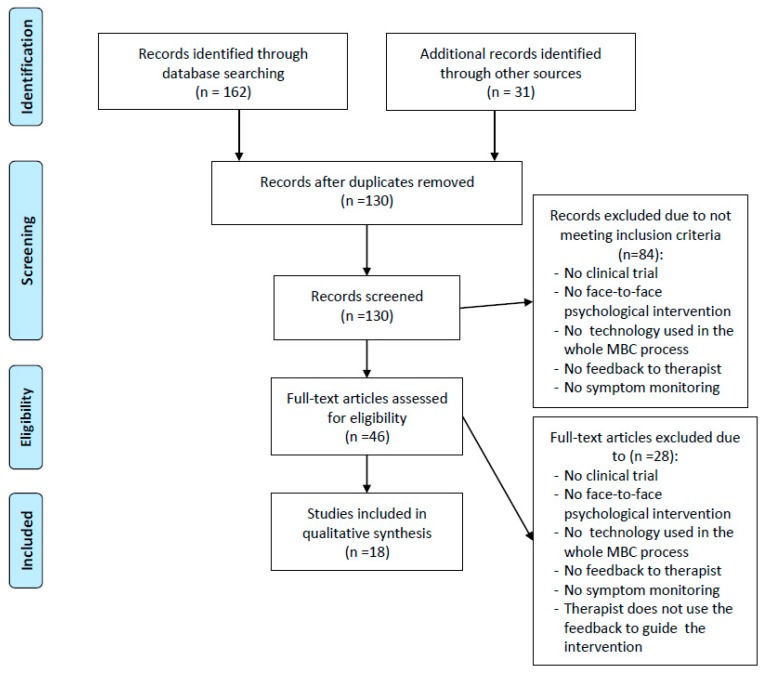
Flow diagram of study selection following PRISMA guidelines [26].

**Table 1 ijerph-17-03170-t001:** Characteristics of the included studies.

Reference	Country	Setting	Psychological Disorder	Sample size	Study Design	Type of Psychological Intervention
[30]	California (USA)	Specialist inpatient treatment center	PTSD	27 Tx	SG (feasibility study)	Residential group psychotherapy (4–5 months)
[31]	Pittsburgh (USA)	Outpatient Psychiatric Institute and Clinic	Anxiety disorders (children from 9 to 14 years old)	9 Tx (3 received 16-sessions CBT, 6 received 8-sessions Brief CBT)	SG (feasibility pilot trial)	“Coping Cat”: 16 CBT sessions“Brief Coping Cat”: 8 CBT sessions
[32]	Evanston, Illinois (USA)	Outpatient clinic at the Family Institute at Northwestern University	Couple problems	1 Tx	Case study	Four Couple Multisystemic Psychotherapy sessions
[33]	California University: Berkeley (USA)	Outpatient behavioural health clinic	Depression	85 (40 cont. + 45 Tx)	nRCT (not blinded)	16 weeks of weekly group CBT therapy
[34]	California (USA)	4 Outpatient Early Psychosis clinics	Psychotic disorder	61 Tx	SG (feasibility study)	Early Psychosis Program (up to 5 months)
[35]	New York (USA)	Outpatient primary care	Substance use	240 (83 cont. + 77 Tx + 80 Tx HealthCall)	Three-arm RCT (1:1:1 allocation ratio)	Brief (25–30 min) individual (motivational interview) psychoeducation (3 sessions: every 30 days)
[36]	California University (USA)	Outpatient specialist clinic	Psychotic disorder (adolescent and young)	76 Tx	SG (feasibility pilot trial)	Early Psychosis Program (minimum 3 months)
[37]	Washington University (USA)	Outpatient primary care clinic affiliated with the Washington University	Depression and Anxiety	17 Tx	SG (feasibility and acceptability pilot study)	Collaborative care program (over 6 months)
[38]	Melbourne (Australia)	Outpatient Specialist Voices Clinic and clinical services	Schizophrenia	34 (17 cont. + 17 Tx)	A single-blind, parallel group, pilot RCT (1:1 allocation ratio)	Brief CBT (four in-person therapy sessions) + EMA + EMI
[39]	Melbourne (Australia)	Outpatient Specialist Voices clinic	Schizophrenia	1 Tx	Case study	Brief CBT (four in-person therapy sessions) + EMA+ EMI
[40]	Perth (Australia)	Private inpatients and day-patients psychiatric hospital	Mood and Anxiety	1308 (408 Tx Fb +, 439 nFb + 461 cont.)	nRCT	10 days of intensive CBT group
[41]	Leeds (England)	Outpatient clinic	Depression and Anxiety	594 (349 cont. + 245 Tx)	Quasi-experimental pre-post study	Low-intensity guided self-help CBT or High-intensity CBT, interpersonal psychotherapy and counselling
[42]	England	8 outpatient clinics	Depression and Anxiety	2233 (1057 cont. + 1176 Tx)	Multisite, open-label, cluster RCT	Low-intensity guided self-help CBT or High-intensity CBT, interpersonal psychotherapy and counselling
[43]	Oxford (UK)	Outpatient specialist mental health	Bipolar disorder	19 Tx	SG (pre-post)	five sessions of psychoeducational intervention (FIMM) + pharmacotherapy
[44]	Salzburg (Austria)	Inpatient and a day-treatment clinic	Mood disorders; Psychoactive substance use mental disorders; Schizophrenia, schizotypal, and delusional disorders; Neurotic, stress-related, and somatoform disorders; Personality disorders, and others	151 Tx	SG (feasibility pilot trial)	Psychotherapy (8 weeks in the day-treatment clinic and 12 weeks in the inpatient clinic)
[45]	Salzburg (Austria)	Outpatient clinic	Bulimia nervosa	1 Tx	Case study	six Rogerian person-centred psychotherapy sessions
[46]	Thessaloniki (Greece)	Alzheimer day care centre (at home)	Dementia	4 Tx	Case study	15 Individual psychotherapy sessions (psychosocial intervention, CBT, relaxation techniques, etc.)
[47]	The Netherlands	Outpatient mental health institutions or private practices	Mood disorder, Adjustment disorder, Anxiety disorder, Relational problems, Personality disorders, and others	475 (159 Tx FbT; 172 Tx FbTP; 144 cont. nFb)	RCT	Long and short psychotherapy (CBT, client-centered, psychodynamics)

Note: Cont., control group; Tx, treatment group; nRCT, non-randomized controlled trial; CBT, Cognitive-Behavioural Therapy; RCT, randomized controlled trial; EMA, ecological momentary assessment; EMI, ecological momentary intervention; FbT, feedback to the therapist; FbTP, feedback to the therapist and the patient; Fb, feedback; nFb, no-feedback; FIMM, Facilitated Integrated Mood Management; SG, single group design.

**Table 2 ijerph-17-03170-t002:** Measurement-based care characteristics.

Reference	Assessment Frequency and Setting	Primary Outcome Measures	Feedback to	Feedback Frequency and Setting	Type of Technology Used
[30]	Daily at a random time	Adapted questionnaire from Symptom Checklist-6, the BriefCOPE and Beck Depression Inventory-II	T and P	P: T regularly shared P progress in order to incorporate strategies in therapy sessions and treatment plan.T: They got patient information several times a week in a graph format to discuss with them during sessions, to encourage them and monitor them.	EMA and Text messages
[31]	Daily questions about recent emotional events (e.g., emotions, scenario, somatic symptoms, automatic thoughts) + answers on demand by the participant	Skills entries and satisfaction with the treatment.	P and T	P: They received personalized feedback from therapists.T: Information and graphs from the portal about patients’ progress were discussed in weekly CBT sessions with the patients.	Smartphone app: SmartCAT app + SmartCAT therapist portal..
[32]	Online before every session	STIC: set of questionnaires	T and clinicians stakeholders	T: On-demand graphs of patient progress were provided to the therapist through STIC	STIC online
[33]	Once daily mood monitoring messages at random between 8 a.m. and 8 p.m.	Attendance to therapy, duration of therapy and PHQ-9	P and T	P: They received feedback about their mood responsesT: T reviewed information from an online dashboard were patient progress is shown. T can periodically review the graphs, identify key aspects and address any important event during or between sessions.	Automated text messages and Web-based platform (HealthySMS)
[34]	Daily and weekly surveys (between 5 p.m. and 10:30 p.m.)	Mood, medication use, socialization and conflict	P and T	P: They discussed their feedback with the T at every session.T: They reviewed and discussed patient information (plots of symptoms over time, etc.) on the dashboard with P during sessions and between sessions.	Smartphone app: RealLife Exp + web-based platform
[35]	Once daily call (HealthCall) for self- monitoring	Primary drug use (frequency and amount), use of other drugs, medication adherence, and mood	P and T	P: They received the feedback at 30 and 60 days, where their information was discussed with the T.T: At 30 and 60 days, T discussed with P the generated graphs based on HealthCall about their drug use, moods and health behaviors.	Phone IVR system
[36]	Daily surveys (at 5 p.m. until 11:55 p.m.), weekly surveys (Sundays at 10 a.m. until Monday 11:55) and monthly in-person psychosocial assessments with research staff	Daily surveys assessing mood, medication adherence, and social interaction, weekly surveys assessing symptoms, sleep, and medication adherence	T	T: They received alerts from the dashboard when P scores were clinically significant and took the proper decisions according to the patient demand.	Ginger.io (software) = Smartphone app + Clinician dashboard
[37]	3/4 times daily, weekly; 8/12 weeks	PHQ-9 and GAD-7	P and T	P: P received notifications about their progress in the app, becoming more aware of their symptoms.T: T reviewed patient-reported information via an online dashboard and visualized patient progress graphs.	Smartphone app + online platform
[38]	Session 1 and 2: 10 daily evening EMA for 6 days. Session 3 and 4: 8 evening daily EMA (monitor changes in voices and coping strategies)	SEPS, PSYRATS-AH, and DASS-21	P and T	P: In session 2, P received a summary sheet with their EMA progress.T: In session 2, EMA feedback was discussed with the P in order to guarantee understanding, detect predictors and avoid causation.	Smartphone app: MovisensXS + web-based platform
[39]	Session 1 and 2: 10 daily evening EMA for 6 days. Session 3 and 4: 8 evening daily EMA (monitor changes in voices and coping strategies)	SEPS, PSYRATS-AH, and DASS-21	P and T	P: In session 2, P received a summary sheet with their EMA progress.T: In session 2, EMA feedback was discussed with the P in order to guarantee understanding, detect predictors and avoid causation.	Smartphone app: RealLife Exp + web-based platform
[40]	Daily self-reported measures of well-being	Well-being (WHO-5)	P and T	P: They received routinely individualized information about their progress in group discussion with the therapist.T: T received daily automatic plots of each patient’s outcomes within trajectories.	Touch-screen technology in therapy rooms
[41]	Weekly (session-by-session)	PHQ-9 and GAD-7	P and T	P: They received their feedback in session with the therapist, where the information was reviewed, discussed and used to guide the treatment plan.T: They had access to patient progress graphs and response curves from the monitoring system, and they were warned automatically when a P was not-on-track.	Computer PCMIS
[42]	Weekly (session-by-session)	PHQ-9 and GAD-7	P and T	P: They received their feedback in session with the therapist, where the information was reviewed, discussed and used to guide the treatment plan.T: They had access to patient progress graphs and response curves from the monitoring system, and they were warned automatically when a P was not-on-track.	Computer PCMIS
[43]	Twice a day (only during psychoeducation sessions) and once a week	Daily: mood and sleepWeekly: QIDS, ASRM and mood management strategies questionnaire	P and T	P: P received their feedback at every session with the T.T: T reviewed patient progress from daily mood rating and weekly scales from the previous week and discussed with the P the relationship between his/her mood changes and stressors.	Phone text messages or e-mails (True Colours mood monitoring system)
[44]	Daily process monitoring (during evenings).	TPQ	T	T: On demand. Feedback was used for individualizing therapeutic decisions.	SNS
[45]	Daily measures of psychotherapy process. Weekly measure of therapy outcome.	IEQ daily, weekly SCL-90 (Bulimia)	P, T and researchers	P: They viewed their progress and estimated their moods and symptoms during the past day.T: They had access to P data from the system in order to adapt the intervention delivered.	Smartphone: DynAMo web app
[46]	Daily monitoring	Sleep patterns, physical activity, and activities of daily living	P, T and caregivers	P and caregiver: They could see a proportionate share of the information adapted to their needs.T: Information collected was available at all times in order to design personalized interventions.	Tablet app (assistive technology: wearable, sleep, movement, presence sensors)
[47]	Once a week just before therapy session (at waiting room)	OQ-45	P and T, or only T	P: P can access the feedback via email or into their portal system.T: T could access the feedback via email or in their portal system and could discuss the feedback information (progress charts and a message) with the P based on the OQ-45 patient’s scores.	Computer: Web-based monitoring app

Note: PHQ-9, Patient Health Questionnaire-9; P, patient; T, therapist; GAD-7, Generalised Anxiety Disorder-7; EMA, Ecological Momentary Assessment; SEPS, Subjective Experiences of Psychosis Scale; PSYRATS-AH, Auditory Hallucinations subscale of the Psychotic Symptom Rating Scales; DASS-21, Depression Anxiety Stress Scale; OQ-45, Outcome Questionnaire; IEQ, Intersession Experience Questionnaire; SCL-90, Symptom Checklist-90; QIDS, Quick Inventory of Depressive Symptomatology; ASRM, Altman Self Rating Mania Scale; WHO-5, World Health Organization’s Wellbeing Index; STIC, Systemic Therapy Inventory of Change System; CBT, Cognitive Behavioral Therapy; TPQ, Therapy Process Questionnaire; IVR, interactive voice response; App, mobile application; STIC, Systemic Therapy Inventory of Change System; SNS, Synergetic Navigation System; PCMIS, Patient Case Management Information System.

**Table 3 ijerph-17-03170-t003:** Usability, acceptability, and effectiveness of technology-supported measurement-based care.

Reference	Sample Size	Feasibility of Technology	Clinical Effectiveness
[30]	27 Tx	Monthly: 92.7%	NA
[31]	9 Tx	Completion rate was 82.8%. Patients reported the app being easy to use. All parents report treatment satisfaction and would recommend the program.	NA
[33]	85 (40 cont. + 45 Tx)	NA	Technology-supported MBC significantly increased treatment adherence (median of 13.5 weeks before dropping out) compared to traditional CBT (median of 3 weeks before dropping out). Effect sizes of technology-supported MBC CBT on depressive symptoms’ severity (z = −5.80) were larger than for traditional CBT (z = −3.12), but differences were not significant
[34]	61 Tx	Moderate survey completion (daily = 40%; weekly = 39%). In general, both T (66%) and P (85%) reported they would continue using the app as part of the treatment.	NA
[35]	240 (83 cont. + 77 Tx + 80 Tx)	HealthCall shows a great retention rate and response rate (64.1%), supporting feasibility, patient acceptability and generalizability.	At 12-month follow-up, reductions in non-injection drug use were comparable in traditional and technology-supported MBC motivational interviewing and superior than in the control condition. In the subset of patients with drug dependence, drug use was significantly lower in the technology-supported MBC condition at 12 months post-treatment. At 60 days, treatment retention in the technology-supported MBC group (88.8%) was superior than in the motivational intervention only condition (81.8%) and the control condition (78.3%)
[36]	76 Tx	Feasibility and acceptability of the smartphone app as an adjunct treatment tool is supported by the high response rate sate (weekly surveys: 77%; daily surveys: 69%)	NA
[37]	17 Tx	The feasibility and acceptability of the mobile platform is supported by the high early response rate (weekly = 88%).	NA
[38]	34 (17 cont. + 17 Tx)	High completion rates (74%) of EMA questionnaires and good satisfaction of participants support the feasibility and acceptability of the study, respectively.	Compared with the usual treatment, the technology-supported MBC treatment resulted in large improvements in confidence in coping with voices (Hedges g = 1.45) and medium improvements in understanding of voices (Hedges g = 0.61) and in psychotic symptoms (Hedges g = 0.51). Both groups showed similar changes in the impact of psychosis.
[40]	1308 (408 Tx Fb +, 439 nFb + 461 cont.)	High rates of touch-screen questionnaire completion (over 90%).	Technology-supported MBC for NOT patients was more effective than traditional CBT or monitoring without feedback in reducing depressive symptoms and the impact of emotions on functioning, as well as on increasing vitality. By contrast, changes in well-being, anxiety, and stress were comparable across conditions.
[41]	594 (349 cont. + 245 Tx)	MBC technology was generally acceptable and feasible to integrate in routine practice.	Technology-supported MBC achieved comparable reductions in depression and anxiety compared to controls, but with significantly less time (adjusted mean = 10.25, SE = 0.45 vs. adjusted mean = 6.59, SE = 0.51) and cost (between £65.88 and £129.20 cost reductions per treatment). Cases in the control condition were twice as likely to become not-on-track patients compared to those in the technology-supported MBC.
[42]	2233 (1057 cont. + 1176 Tx)	NA	NOT patients in the technology-enhanced MBC condition obtained significantly larger reductions in depressive (*d* = 0.23) and anxiety symptom severity (*d* = 0.19), as well as improved work and social adjustment (*d* = 0.19) compared with active controls (traditional CBT).
[43]	19 Tx	High response rate (daily = 81%, weekly = 88%)	NA
[44]	151 Tx	High average compliance rates (78.3%) and low average missing data rates (13%) amongst the inpatients support the feasibility.	NA
[47]	475 (159 Tx FbT; 172 Tx FbTP; 144 cont. nFb)	NA	In short-term interventions (less than 35 weeks), receiving feedback was protective of negative outcomes in NOT cases (*d* = 1.28). No significant differences between conditions were found for on-track patients, but there was a trend for the technology-supported MBC group to be more effective (*d* = 0.24 at 35 weeks and *d* = 0.29 at 78 weeks) and to have lower deterioration rates (*z* = 1.3), especially when feedback was provided to both patient and therapist.

Note: Cont., Control Group; Tx, Treatment Group; FbT, Feedback to The Therapist; FbTP, Feedback to The Therapist and The Patient; Fb, Feedback; nFb, No Feedback; NA, Not Applicable/Not Specified; IVR, Interactive Voice Response; App, Mobile Application; EMA, Ecological Momentary Assessment; MBC, Measurement-Based Care; NOT, Not-on-Track patients; T, Therapist; P, Patient. The feasibility and effectiveness reports are not provided for case studies due to the reduced number of patients (*n* ≤ 4).

**Table 4 ijerph-17-03170-t004:** Quality assessment of case studies.

	[32]	[39]	[45]	[46]
1. Was the study question or objective clearly stated?	Yes	Yes	Yes	Yes
2. Was the study population clearly and fully described, including a case definition?	No	Yes	No	Yes
3. Were the cases consecutive?	NA	NA	NA	No
4. Were the subjects comparable?	NA	NA	No	Yes
5. Was the intervention clearly described?	Yes	Yes	Yes	Yes
6. Were the outcome measures clearly defined, valid, reliable, and implemented consistently across all study participants?	Yes	Yes	Yes	Yes
7. Was the length of follow-up adequate?	NR	No	No	NR
8. Were the statistical methods well-described?	Yes	Yes	Yes	Yes
9. Were the results well-described?	Yes	Yes	Yes	Yes
Total score (maximum 9 points)	5	6	5	7

Note: CD, Cannot Determine; NA, Not Applicable; NR, Not Reported.

**Table 5 ijerph-17-03170-t005:** Quality assessment of before–after studies.

	[41]	[43]
1. Was the study question or objective clearly stated?	Yes	Yes
2. Were eligibility/selection criteria for the study population prespecified and clearly described?	Yes	Yes
3. Were the participants in the study representative of those who would be eligible for the test/service/intervention in the general or clinical population of interest?	NR	Yes
4. Were all eligible participants that met the prespecified entry criteria enrolled?	NR	No
5. Was the sample size sufficiently large to provide confidence in the findings?	Yes	No
6. Was the test/service/intervention clearly described and delivered consistently across the study population?	Yes	Yes
7. Were the outcome measures prespecified, clearly defined, valid, reliable, and assessed consistently across all study participants?	Yes	Yes
8. Were the people assessing the outcomes blinded to the participants’ exposures/interventions?	No	NA
9. Was the loss to follow-up after baseline 20% or less? Were those lost to follow-up accounted for in the analysis?	Yes/Yes	Yes/Yes
10. Did the statistical methods examine changes in outcome measures from before to after the intervention? Were statistical tests done that provided p values for the pre-to-post changes?	Yes	Yes
11. Were outcome measures of interest taken multiple times before the intervention and multiple times after the intervention (i.e., did they use an interrupted time-series design)?	Yes	Yes
12. If the intervention was conducted at a group level (e.g., a whole hospital, a community, etc.) did the statistical analysis take into account the use of individual-level data to determine effects at the group level?	Yes	Yes
Total score (maximum 12 points)	9	9

Note: CD, Cannot Determine; NA, Not Applicable; NR, Not Reported.

**Table 6 ijerph-17-03170-t006:** Quality assessment of observational cohort and cross-sectional studies.

	[30]	[31]	[34]	[36]	[37]	[44]
1. Was the research question or objective in this paper clearly stated?	Yes	Yes	Yes	Yes	Yes	Yes
2. Was the study population clearly specified and defined?	Yes	Yes	Yes	Yes	Yes	Yes
3. Was the participation rate of eligible persons at least 50%?	NR	NR	Yes	Yes	No	NR
4. Were all the subjects selected or recruited from the same or similar populations? Were inclusion and exclusion criteria for being in the study prespecified and applied uniformly to all participants?	Yes	Yes	Yes	Yes	Yes	Yes
5. Was a sample size justification, power description, or variance and effect estimates provided?	NR	NR	NR	NR	NR	NR
6. For the analyses in this paper, were the exposure(s) of interest measured prior to the outcome(s) being measured?	Yes	Yes	Yes	NA	Yes	Yes
7. Was the timeframe sufficient so that one could reasonably expect to see an association between exposure and outcome if it existed?	NA	NA	NA	NA	NA	NA
8. For exposures that can vary in amount or level, did the study examine different levels of the exposure as related to the outcome (e.g., categories of exposure, or exposure measured as continuous variable)?	NA	NA	NA	NA	NA	NA
9. Were the exposure measures clearly defined, valid, reliable, and implemented consistently across all study participants?	Yes	Yes	Yes	Yes	Yes	Yes
10. Was the exposure(s) assessed more than once over time?	Yes	Yes	Yes	Yes	Yes	Yes
11. Were the outcome measures clearly defined, valid, reliable, and implemented consistently across all study participants?	Yes	Yes	Yes	Yes	Yes	Yes
12. Were the outcome assessors blinded to the exposure status of participants?	NA	NA	NA	NA	NA	NA
13. Was loss to follow-up after baseline 20% or less?	Yes	Yes	No	No	No	No
14. Were key potential confounding variables measured for their impact on the relationship between exposure(s) and outcome(s)?	NA	NA	NA	NA	NA	NA
Total score (maximum 14 points)	8	8	8	7	7	7

Note: CD, Cannot Determine; NA, Not Applicable; NR, Not Reported.

**Table 7 ijerph-17-03170-t007:** Quality assessment of controlled intervention studies.

	[33]	[35]	[38]	[40]	[42]	[47]
1. Was the study described as randomized, a randomized trial, a randomized clinical trial, or an RCT?	No	Yes	Yes	No	Yes	Yes
2. Was the method of randomization adequate (i.e., use of randomly generated assignment)?	NA	Yes	Yes	NA	Yes	Yes
3. Was the treatment allocation concealed (so that assignments could not be predicted)?	NA	Yes	Yes	NA	Yes	Yes
4. Were study participants and providers blinded to treatment group assignment?	No	No	No	No	No	No
5. Were the people assessing the outcomes blinded to the participants’ group assignments?	No	NA	Yes	No	No	NA
6. Were the groups similar at baseline on important characteristics that could affect outcomes (e.g., demographics, risk factors, co-morbid conditions)?	Yes	Yes	Yes	Yes	Yes	No
7. Was the overall drop-out rate from the study at endpoint 20% or lower of the number allocated to treatment?	Yes	Yes	Yes	NR	No	No
8. Was the differential drop-out rate (between treatment groups) at endpoint 15 percentage points or lower?	Yes	Yes	Yes	NR	Yes	Yes
9. Was there high adherence to the intervention protocols for each treatment group?	Yes	Yes	Yes	NR	Yes	Yes
10. Were other interventions avoided or similar in the groups (e.g., similar background treatments)?	Yes	NR	Yes	NR	Yes	NR
11. Were outcomes assessed using valid and reliable measures, implemented consistently across all study participants?	Yes	Yes	Yes	No	Yes	Yes
12. Did the authors report that the sample size was sufficiently large to be able to detect a difference in the main outcome between groups with at least 80% power?	No	Yes	No	NR	Yes	Yes
13. Were outcomes reported or subgroups analyzed prespecified (i.e., identified before analyses were conducted)?	Yes	Yes	Yes	Yes	Yes	Yes
14. Were all randomized participants analyzed in the group to which they were originally assigned, i.e., did they use an intention-to-treat analysis?	Yes	No	Yes	NR	Yes	Yes
Total score (maximum 14 points)	8	10	12	2	11	9

Note: CD, Cannot Determine; NA, Not Applicable; NR, Not Reported.

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
