# Peer review of "How Are Information and Communication Technologies Supporting Routine Outcome Monitoring and Measurement-Based Care in Psychotherapy? A Systematic Review"

_ijerph, 2020, doi:10.3390/ijerph17093170_

Round 1
Reviewer 1 Report
Thank you for the opportunity to review this manuscript. This article was a systematic review of information and communication technologies supporting ROM and MBC. The information provided in this study is timely and significant given the current global pandemic and the significance of providing psychotherapy that is supplemented with technology.
Comment 1: The authors could better explain why the three databases (PubMed, PsycINFO, and SCOPUS) were selected and why others were not selected. This will provide a better rationale of why the authors were only able to retrieve 162 articles from their initial database searching.
Comment 2: The authors mentioned Appendix 1, but I could not seem to locate this information in the manuscript. The search strategy will be helpful to better understand the systematic process that was taken by the authors.
Comment 3: I would recommend including citations throughout the narrative of the results. For instance, in addition to include the n for the outpatient settings, the authors should also consider including the n per site (i.e., mental health services, specialist clinics, etc.).
Comment 4: In addition to adding the n, it will be helpful if the authors included the numbered citations with the narrative. For instance, the authors may cite the numbered references of the articles that took place in Australia, England, Austria, etc.
Comment 5: In line 212-213, the authors indicated “Most studies (n=9) monitored their patients daily. The remaining studies monitored their patients weekly (n=4), before every therapy session, or both, daily and weekly (n=5).” It was unclear what was meant by the studies that monitored their patients both daily and weekly. Please clarify.
Comment 6: For the section on Technology feasibility, citations are needed for the studies that were cited with high average response rates.
Comment 7: For the section on clinical effectiveness, the authors did not provide an explanation of the studies that did not show any difference in outcomes between groups. This would seem particularly important to discuss given the results of studies 33, 47, 51, 46, and 35.
Comment 8: The tables are very comprehensive and provide a concise overview of each study. However, it was unclear why the references were numbered in such a manner. One suggestion is to number the studies in the order they are presented. I was also confused as to how the tables were organized. Initially, I thought the tables were based on country. After careful review, I realized this was not the case. I recommend the authors re-organize the tables to increase the meaningfulness of each table.
Comment 9: Please include the total score for each study included in the quality assessment tables. This will help the readers to make quick comparisons between the different studies.
Comment 10: For the discussion, the authors should also discuss the limitations of only using three databases in their systematic review. Furthermore, this systematic review is limited to the interpretations of the authors who conducted the systematic review.
Overall, interesting findings with significant content.
Editorial:
Line 39: I would consider another word for “abandon”. There may be contextual factors that prevents a person from continuing on with psychotherapy rather than “abandoning” treatment.
Line 74: edit “to therapists has demonstrated to reduce”
3.4. Risk of bias assessment: edit “sessions or it these were not reported”
Discussion: Consider word choice of “investigation”. I would suggest “the present study”.
Discussion: edit “In addition to the previous, it is also…”
Reviewer 2 Report
The authors were set out to see whether the published literature supports the feasibility and effectiveness of digital health interventions in psychiatric applications.
My comments and suggestions for changes are the following:
Line 101: use past tense --> have conducted
Line 103: use past tense --> 'have investigated'
Line 128: 'routine' instead of routinely
Line 185: delete 'of them'
Line 187 following: in my view, case studies with 1 or very few patients should be excluded. But if they are included in the review, they should at least not be included in the effectiveness summary.
Line 262-264:it is unclear where these data come from (is a reference to a table missing?)
Line 270:explain abbreviation 'NOT'
Line 269-270:The text say "the use of technology-supported MBC appears to significantly reduce symptom severity in patients at risk of poor response to treatment". This is not what I see in Table 3, where it is indicated that several studies showed no clinical effect (e.g. references #33, 51, 46, 35 ...)
Line 272:'significant' instead of insignificantly
Table 3:Table 1 indicates that the references 51, 36, 12 are case reports with N=1 and 41 with N=4. The results report 'completion rate of 33%' (#51), 'Compliance rates (80%) and missing data rates (10%) for outpatients' (#36), 'Positive improvement on patients’ symptoms in the experimental group' (#41). This is a bit strange and misleading with only 1 (or 4) patient. Actually, the result table should include a column with the N (experimental vs control).
Table 4: The reference numbering seems inconsistent. In Table 1, #51, 36, 12 and 41 are listed as case studies. #52 is not listed in Table 1
Table 4-7: The legend of the tables contains the following sentence 'We included the first authors’ name only due to space limitations' but I cannot see any author names.
Limitations, Page 17: could another limitation be that studies or pilots or cases, where ICT wasn't feasible or didn't add any value, were not published?
Limitations, page 17, last sentence: I am not so sure that RCTs allow for establishment of causal relationships. I thought, the strength of RCTs is a higher internal validity and therefore, a higher robustness of the evidence indicating a (causal) relationship.
Line :
Round 2
Reviewer 2 Report
I would like tot thank the author for their fast response and the changes which were implemented. It was a bit confusing that long passages were marked as changed - although they were not really changed from the previous version.
For table 3, I would still prefer to the the N of the studies in the table as it is easier to understand without having to go back between tables. But I leave the decision to the authors.
Author Response
Dear reviewer,
thanks a lot for the words. We are glad we addressed the concerns raised. We agree that Table 3 is clearer with sample sizes in it. We have added that and made the corresponding changes to the Table note.
Regards,
Carlos